# Iron Status in Pregnant Women in Latvia: An Epidemiological, Cross-Sectional, Multicenter Study According to WHO and UK Criteria

**DOI:** 10.3390/medicina58070955

**Published:** 2022-07-19

**Authors:** Roberta Rezgale, Iveta Pudule, Vinita Cauce, Kristine Klaramunta Antila, Violeta Bule, Gunta Lazdane, Dace Rezeberga, Laila Meija

**Affiliations:** 1Faculty of Medicine, Rīga Stradiņš University, 16 Dzirciema Street, LV-1007 Rīga, Latvia; roberta.rezgale@rsu.lv (R.R.); vinita.cauce@rsu.lv (V.C.); kristine.claramunt@gmail.com (K.K.A.); violeta.bule@rsu.lv (V.B.); gunta.lazdane@rsu.lv (G.L.); dace.rezeberga@rsu.lv (D.R.); 2Centre for Disease Prevention and Control of Latvia, 22 Duntes Street, LV-1005 Rīga, Latvia; iveta.pudule@spkc.gov.lv; 3Riga East Clinical University Hospital, 2 Hipokrāta Street, LV-1038 Rīga, Latvia; 4Riga Maternity Hospital, 45 Miera Street, LV-1013 Rīga, Latvia; 5Pauls Stradins Clinical University Hospital, 13 Pilsoņu Street, LV-1002 Rīga, Latvia

**Keywords:** iron deficiency, anaemia, pregnancy

## Abstract

*Background and Objectives*: During pregnancy, iron deficiency anaemia is a common problem associated with health risks for both the mother and her foetus/infant. This study aimed to investigate the prevalence of iron deficiency, iron deficiency anaemia, and related dietary patterns in pregnant women in Latvia. *Materials and Methods*: This cross-sectional, multicentre study included pregnancy data from 974 women. The sample selection was based on the stratification principle (population of women of childbearing age in regions of Latvia). Maternal demographic details, anthropometric measurements, iron status, dietary patterns, and supplementation information were obtained from maternal files and during interviews held in eight outpatient departments of medical institutions and maternity departments. The prevalence was assessed. Chi-square tests and logistic regression were used to identify associations between iron deficiency and sociodemographic characteristics, dietary patterns, and iron supplement intake during pregnancy. The criterion used for the diagnosis of iron deficiency anaemia is a Hb level < 110 g/L in the 1st and 3rd trimesters and <105 g/L during the 2nd trimester as recommended by the WHO. However, the UK guideline was used for borderline iron deficiency, which is an SF level < 30 μg/L in all trimesters. *Results*: The observed prevalence of anaemia was 2.8% in the first trimester, 7.9% in the second trimester, and 27.0% in the third trimester. The prevalence of iron deficiency was 46.7% in the first trimester, 78.1% in the second trimester, and 91.7% in the third trimester. No associations with dietary patterns were found. Single women had 1.85 times the odds (95% CI 1.07 to 3.18) of being anaemic than married women. *Conclusions*: Iron deficiency affects a large proportion of pregnant women in Latvia in all trimesters, with iron deficiency anaemia affecting pregnant women in the third trimester. Monitoring and intervention should be performed in a timely and more targeted manner.

## 1. Introduction

Due to increased iron requirements, women during pregnancy have the highest risk of iron deficiency (ID) and related anaemia [1,2]. The increased foetal demand for iron during pregnancy is met mostly by maternal iron stores, which results in pregnant women being at higher risk of developing ID and ID anaemia [2].

Maternal ID with or without anaemia during pregnancy is reported to negatively affect the mother and the foetus/newborn [3,4]. Mothers with ID often experience increased levels of fatigue and reduced physical and mental performance [5]. Severe maternal ID anaemia is associated with an increased risk of caesarean delivery, transfusion, perinatal bleeding, pre-eclampsia, placental abruption, poor maternal thyroid status, poor wound healing, cardiac failure, and even death [6,7,8,9]. The adverse effects of ID for the foetus/newborn are increased risks of adverse birth outcomes such as preterm birth, low birth weight, low neonatal iron stores at birth, and poor overall infant health [10,11]. Accumulative data suggest that the developing foetal brain is susceptible to iron insufficiency in utero, and low iron stores at birth have been associated with poorer cognitive, motor, social-emotional, and neurophysiologic development in infants [12,13,14].

It is important to mention that there is no common global agreement on the diagnostic criteria for ID and ID anaemia in pregnant women. Specifically, it is difficult to establish a cut-off point that takes into account the many physiological changes that occur during pregnancy, including changes in blood composition, haemodynamics, inflammatory status, and hormones [15]. The following WHO parameters are used worldwide to diagnose ID anaemia: haemoglobin (Hb) < 110 g/L in the 1st and 3rd trimesters and Hb < 105 g/L in the 2nd trimester [16]. The UK guidelines on the management of ID anaemia in pregnancy include a decrease in Hb levels in the 3rd trimester to <105 g/L [17]. On the other hand, serum ferritin (SF) is the most commonly used indicator of iron status. The WHO guidelines define ID in pregnant women as borderline when the SF value is 15 µg/L but include remarks that there is limited evidence available on establishing cut-off points by trimester [15], whereas the UK guidelines note that SF levels < 30 µg/L indicate ID throughout the pregnancy [17]. Obstetricians, gynaecologists, and other health professionals involved in antenatal care in Latvia use the WHO criteria for the diagnosis of ID anaemia and the UK criteria for the diagnosis of ID during pregnancy [18].

Iron status during pregnancy is influenced by dietary intake of iron. Eating patterns, including enhancers and inhibitors of iron absorption, can notably impact iron status [19].

Although ID in pregnancy is identifiable, treatable, and possibly preventable with iron supplementation, there is a difference regarding iron supplementation in pregnancy among developed countries. In several countries, iron supplements of 30–60 mg daily are routinely administered to pregnant women [20]. Some research claims that ignoring the body’s adaptation processes during pregnancy and using inadequate doses of iron supplements can lead to a variety of health problems ranging from endometriosis to pre-eclampsia [21].

In other countries, such as the US, UK, and Latvia, the routine use of iron supplements is not recommended, and the need for iron supplements is assessed on an individual basis [22,23]. Many pregnant women use multivitamin supplements containing iron without a prescription from a physician.

According to the World Health Organization (WHO), the prevalence of anaemia was 24.5% among pregnant women in Europe in 2011 [1] and 23.2% among pregnant women in Latvia in 2019 [24]. However, the latter report does not specify the type of anaemia. On the other hand, although ID may also have serious health risks, existing newborn register data are insufficient or missing. Despite decades of public health interventions, ID and ID anaemia remain a significant problem during pregnancy in Latvia. The Centre for Disease Prevention and Control (CDPC) of Latvia reports that the prevalence of ID anaemia in pregnant women has not decreased in recent years [25]. Thus, this study aims to assess the prevalence of ID and ID anaemia, as well as related eating patterns among pregnant women in Latvia.

## 2. Materials and Methods

### 2.1. Study Design and Participants

The study was conducted by Riga Stradiņš University in cooperation with the CDPC of Latvia and the Country Office of the WHO in Latvia. This cross-sectional, multicentre study was conducted between August 2017 and October 2019 and included pregnancy data from pregnant and postpartum women up to the 7th day after delivery, all of whom maintained their routine diets. The study took place in 8 maternity outpatient clinics and maternity departments in hospitals in all regions of Latvia.

The study population included pregnant or up to 7th-day postpartum women who had resided in Latvia for more than one year. In contrast, exclusion criteria were being less than 18 years; multiple pregnancies; place of residence in another country during pregnancy; or previous or existing health disorders, including an eating disorder, diabetes, celiac disease, short bowel syndrome, Crohn’s disease, or ulcerative colitis.

The sample size was calculated to be representative of the proportion of women in Latvia of reproductive age (15 to 49 years old) with a 3% margin of error, which would have required 1089 participants.

More than one thousand women were screened; however, 18 out of 1091 (0.7%) participants did not meet the inclusion criteria. A total of 970 of 1083 (89.6%) women had Hb data available, whereas 933 of 1083 (86.1%) women had SF data available for analysis. Altogether, a total of 974 from the original 1083 (89.9%) were included in the present analysis, of which 626 were pregnant and 658 were one-week postpartum. The participants were interviewed to obtain dietary and health data, which are described in the “Data Collection” section. The distributions of interviewed participants at one-week postpartum and in different trimesters of pregnancy and the availability of Hb and SF data are presented in Figure 1.

Excluded women did not differ from the other study participants in terms of demographic characteristics such as age, BMI, education, nationality, or Hb level at the first doctor’s visit.

### 2.2. Data Collection

The questionnaire “Dietary Patterns and Influencing Factors of Pregnant Women in Latvia” was adapted from the CDPC of Latvia’s “Health Behaviour among Latvian Adult Population” [26] and was based on WHO expert recommendations. This is a standardized food frequency consumption questionnaire. The questionnaire was prepared in both Latvian and Russian, thus ensuring a complete understanding of the questions being asked and a more comfortable way for the participant to answer. The survey was performed by trained interviewers.

The questionnaire consisted of the following parts: demographic data, medical data, weight, height, health status and healthcare, nutritional patterns, and vitamin and food supplement use. Eating patterns, including eating meat, fish, dairy products, vegetables, and fruits and drinking coffee, were evaluated using a questionnaire [26].

Data on iron status (Hb and SF) were obtained from each participant’s medical records. In Latvia, Hb levels are monitored regularly during all trimesters of pregnancy, with at least one indicator per trimester. Evaluation of SF levels was performed at the first visit and in the 28th week of pregnancy, which is the first week of the third trimester. In the second trimester, SF analyses in the study population were available to less than half of the respondents (Figure 1). If several Hb and SF indicators were available within one trimester, the lowest of the indicators was chosen for further data analysis and the ID and ID anaemia diagnosis.

Information regarding the participant’s weight and height was obtained from the medical records, where weight dynamics were recorded by the medical staff involved in antenatal care and monitoring the weight of the women at each visit. If the information in the medical records was missing, it was replaced with self-reported data (n = 123); for 38 participants, information about weight was missing.

### 2.3. Operational Definitions and Definitions of Terms

#### 2.3.1. Iron Deficiency Anaemia in Pregnancy

According to the WHO criteria, ID anaemia was diagnosed as a Hb level of <110 g/L in the 1st and 3rd trimesters and <105 g/L during the 2nd trimester [18]. According to the “UK guidelines on the management of ID in pregnancy”, anaemia was defined as a Hb concentration < 110 g/L in the 1st trimester and <105 g/L in the 2nd and 3rd trimesters [19].

#### 2.3.2. Iron Deficiency in Pregnancy

The WHO defines ID as an SF level < 15 μg/L only for the 1st trimester [17]. According to the UK guidelines, borderline ID is based on an SF level < 30 μg/L in all trimesters [19].

Therefore, both thresholds were taken into account when analysing the data.

### 2.4. Ethical Considerations

Ethical approval was obtained from the Clinical Research Ethics committee of Pauls Stradiņš Clinical University Hospital (ref: 310717–27L). Written informed consent was obtained from all participants before the study. Participants were provided with written information about the aim and purpose of the study and were offered this information for their records. The study was anonymous. Questionnaires were marked only with code numbers.

### 2.5. Data Analysis

The sample data were screened for the following after collection: accuracy of data entries, identification of missing data, and detection of outliers. Descriptive statistics were obtained, and categorical variables are described as frequencies and percentages. For quantitative data analysis, normal distribution was tested using the Shapiro–Wilk test and graphical methods. If the data were not normally distributed, it was expressed as median (range or interquartile range), and if they were normally distributed, as mean (SD).

Prevalence (CI 95%) of ID and ID anaemia was assessed in all trimesters. Subsequent data were collected in the third trimester, during which women are considered to be the most vulnerable to the development of ID and ID anaemia as maternal and foetal iron requirements increase [24]. Demographic data and BMI were compared for women with or without ID anaemia in all trimesters. For the 3rd trimester, the dietary intake according to Latvian recommendations for pregnant women was compared using chi-square or Fisher–Freeman–Halton tests. Dietary patterns were evaluated according to existing recommendations [25,27,28]. Additionally, logistic regression analysis was performed to evaluate associations between ID anaemia and sociodemographic data in the 3rd trimester.

Hb and SF levels were compared between multivitamin and food supplement users and non-users using the Kruskal–Wallis test. The Spearman correlation was used to analyze the continuous association between ID anaemia and maternal Hb, SF, and multivitamin and iron supplement intake. SPSS version 24.0 was used. A *p*-value <0.05 was considered statistically significant.

## 3. Results

The mean age of the participants was 29.9 years (SD 5.4, from 18.0 to 46.0), and the mean BMI of the participants before pregnancy was 23.8 kg/m^2^ (SD 4.5, from 16.0 to 45.0).

### 3.1. Iron Status in All Trimesters

In the first trimester (*n* = 783), the median Hb level was 126.8 (SD 9.6) g/L. In the second trimester (*n* = 731), the median Hb level was 116.9 (SD 10.7) g/L. In the third trimester (*n* = 788), the median Hb level was 114.7 (SD 9.5) g/L.

In the first trimester, ID anaemia was found in 2.8% of pregnant women (*n* = 22, 95% CI 1.9 to 4.2%) according to the WHO guidelines (Hb < 110 g/L). This prevalence increased to 7.9% (*n* = 58, 95% CI 6.2 to 10.1%) in the second trimester according to the same guidelines (Hb < 105 g/L). The highest prevalence was found in the third trimester with 27.0% of the respondents (*n* = 213, 95% CI 24.1 to 30.2%) when considering the same guidelines (Hb < 110 g/L), but according to its UK analog (Hb < 105 g/L) it was 12.2% (95% CI 10.1 to 14.7%).

On the other hand, according to the WHO classification (SF <15 g/L) [22], ID was present in 18.3% of the pregnant women in the first trimester (*n* = 136; 95% CI 15.7 to 21.3%), but according to the UK guidelines (SF < 30 µg/L) [23], it was 46.7% (*n* = 347; 95% CI 43.1 to 50.3%). In the second trimester, according to the UK guidelines (SF < 30 µg/L), 78.1% women (*n* = 318, 95% CI 73.9 to 81.9%) had ID, whereas 91.7% (*n* = 566, 95% CI 89.3 to 93.7%) had ID in the third trimester.

Table 1 shows the levels of Hb and SF in all trimesters.

Table 2 shows the prevalence of ID and ID anaemia separately according to the two considered guidelines.

### 3.2. Sociodemographic Characteristics

Table 3 presents the maternal demographic characteristics and prenatal BMI classification of women with and without anaemia. Statistically significant associations were found between anaemia and nationality and between anaemia and family status.

### 3.3. Associations between ID Anaemia and Sociodemographic Data

Adjusted by nationality and BMI, the odds of having ID anaemia were higher among single women than among married women. They were also higher among overweight women than among women with a normal BMI adjusted by nationality and marital status (Table 4).

### 3.4. Dietary Iron Intake and Supplement Use

Dietary patterns and iron status in the 3rd trimester.

The median intake of meat in one week was 5.0 (IQR 3.0 to 7.0) portions. The majority of participants (85.2%, *n* = 671) consumed meat according to existing recommendations (≥3 times per week). The median intake of fish was 1 (IQR 1 to 2) portion per week. The recommended fish intake of two portions per week was observed in 15.5% (*n* = 122) of pregnant women. The median intake of dairy products was 2.3 (IQR 1.4 to 3.4) portions per day. Dairy products were consumed 3 to 4 times a day by 20.4% (*n* = 160) of pregnant women, but 15.7% (*n* = 123) of participants exceeded the recommended intake of dairy products. Regular daily consumption of fruits and vegetables was observed in 12.9% (*n* = 103) and 1.3% (*n* = 10) of participants, respectively. There were no statistically significant differences in eating patterns between participants with and without anaemia in terms of the frequency of consuming foods that are sources of iron or influence iron absorption. Sources of iron are meat and fish; fruits and vegetables enhance iron absorption and dairy products and coffee inhibit it (Table 5).

Correlation between multivitamin and iron supplement use with Hb and SF levels.

In the third trimester, multivitamin use during pregnancy was correlated with Hb level (r = 0.112, *p* = 0.001) for 53.7% (*n* = 523) of women but not with the SF level. The median Hb level in this trimester was 116.0 (IQR 109.0 to 121.0) g/L in multivitamin users and 113.0 (IQR 108.0 to 119.0) g/L in non-users (*p* = 0.002).

Iron supplements were used by 55.4% of participants (*n* = 567) during pregnancy. There was a correlation with Hb levels but not SF levels in the second and third trimesters (r = 0.113, *p* = 0.02 and r = 0.077, *p* = 0.030, respectively). The median Hb level in the second trimester was 115.0 (IQR 109.0 to 120.0) g/L in iron supplement users, but 120.0 (IQR 113.3 to 126.0) g/L in non-users. In the third trimester, the median Hb level was 113.0 (IQR 108.0 to 119.0) g/L in iron supplement users but 120.0 (IQR 113.5 to 126.0) g/L in non-users. It was found that 147 (18.6%) participants used iron supplements without having anaemia or ID.

## 4. Discussion

The current study was focused on the prevalence of ID and ID anaemia in pregnant women in Latvia and related eating patterns.

The analysis and interpretation of ID and ID anaemia data in women during pregnancy are complicated by the lack of common diagnostic criteria. There are two parameters to be determined during pregnancy: Hb and SF. However, different diagnostic borderlines are offered by experts and organizations.

Most studies related to the prevalence of anaemia in pregnancy follow the WHO criteria. However, our analysis also took into account the UK guidelines because most gynaecologists and obstetricians in Latvia follow them [20]. UK experts suggested <30 µg/L as the SF cut-off [19] and ID indicator in pregnant women, and it is considered an indication of iron supplementation [20].

In Latvia, seven visits are scheduled for pregnancy monitoring—one in the first trimester, two in the second trimester, and four in the third trimester of pregnancy. Hb levels are monitored regularly at every visit. The evaluation of SF levels is performed at the first visit and in the 28th week of pregnancy, which is the first week of the third trimester [29]. This makes SF data only available in the second trimester for less than half of the respondents in this study and limits the comparison of changes in SF levels between trimesters.

The prevalence of ID anaemia in the first trimester was observed in only 3% of participants, but increased during the course of pregnancy, reaching 27% in the third trimester (according to the WHO criteria) and 12% in the third trimester (according to the UK criteria). These data are in agreement with those of other studies, such as the WHO report of the Global Prevalence of Anaemia in 2011 [1], confirming that anaemia in pregnant women is still widespread and that its prevalence in Latvia has not changed over the last ten years. This is also the case in other Baltic countries such as Estonia where it is 23.3% and Lithuania where it is 21.6% [15]. This is very similar to the 23.2% observed in Latvia in 2019. However, publically available data on iron status is not split between different trimesters of pregnancy in these other countries. Although no data on iron status before pregnancy were available in this survey, the prevalence of ID anaemia in early pregnancy indicates ID prior to pregnancy and that it may be related to preconception care.

Regardless of the SF threshold used to diagnose ID, the prevalence of ID in the current study among pregnant women is high. ID with an SF cut-off of <15 μg/L was observed in 66% of participants in the third trimester, whereas ID with an SF cut-off of <30 μg/L was diagnosed in 92% of pregnant women.

Different studies in Europe have shown variable ID diagnostic parameters as well as different serum marker cut-offs. The reported prevalence of ID varies greatly among different countries, being 40.7% in Germany, which uses soluble transferrin receptor (>3.3 mg/dL) [30], 19% (<12 μg/L) in Switzerland [31], 40% (<15 μg/L) in Belgium, and 60% (<15 μg/L) in France [32], which use SF. There may be several reasons for this, including the preconception and pregnancy diet, iron supplementation, antenatal care, and other factors [33].

The median Hb and SF values were analyzed separately in all trimesters to determine the most critical period of pregnancy with rapid iron changes. The median Hb levels in this study remained within the normal range throughout pregnancy; however, in the third trimester, almost one-third of the participants had anaemia. Between the first and second trimesters, the median Hb level decreased by almost 10 g/L, and between the second and third trimesters, it decreased by approximately 2 g/L. In a British study, the average decrease in Hb between the first and third trimesters was 14 g/L [34], similar to the present study in Latvia.

A similar situation was observed with the average SF values. Specifically, between the first and second trimesters, there was a rapid decrease in SF levels by 20 μg/L and in the second and third trimesters, SF levels continued to decrease by approximately 9 μg/L. It should be noted that the median SF level in the study population in the second trimester had fallen below the cut-off value of SF 30 μg/L, but in the third trimester it had approached the cut-off value of SF 15 μg/L.

Other studies reported similar findings [35,36] starting from week 12 and reaching the lowest level of SF in weeks 35–38. In Latvia, SF levels are monitored at the first visit and at 28 weeks (beginning of the third trimester). Our results show that the risk of ID could be observed much earlier. To effectively prevent ID anaemia, it is important to detect changes in SF levels (<30 μg/L) as soon as possible and to initiate iron supplementation therapy as soon as it is needed.

Our data indicate that the use of multivitamins in the third trimester could help improve iron status; however, iron supplements did not show a benefit. At the same time, a significant number of participants without anaemia or ID used iron supplements. This leads us to think that the present use of multivitamins and iron supplements in Latvia may not be properly targeted and may therefore lack the necessary effects.

Iron status depends not only on total iron intake but also on the absorption and availability of iron in the body [23,37]. Haem iron is absorbed more easily from the main sources, which are meat and fish [38]. Our data show that meat intake among pregnant women in Latvia was within the recommended amount but that there was insufficient fish intake. Vitamin C enhances iron absorption [39]. The main sources of vitamin C are fruits and vegetables, especially citrus fruit, berries, kiwi, broccoli, tomatoes, and peppers [40]. However, the intake of fruit and vegetables was not sufficient in this study. On the other hand, calcium, tea, and coffee inhibit iron absorption [41]. The intake of the main source of calcium—dairy products—was below the recommended level. However, no association between dietary patterns and iron status was found. A possible explanation for this could be that our questionnaire only partly provides information about dietary patterns and that other methods should be used for further detailed investigations of dietary patterns. For example, we did not analyze the association between specific fruits/vegetables and iron status.

The limitation of our study is the absence of Hb and SF levels in some trimesters in some participants. Furthermore, there is no publically available information about how similar the methods for Hb and SF measurements were between institutions. Another limitation is the absence of other laboratory parameters, such as sTfR concentration, and the lack of evaluation of inflammatory status, such as blood CRP determination, taking into account that SF is also a marker of inflammation and that increased SF could represent a response to inflammation. The lower accuracy of self-reported weight data might also affect the study.

## 5. Conclusions

This study demonstrates that ID is prevalent throughout pregnancy and affects a large proportion of pregnant women in the third trimester in Latvia. Monitoring SF during the second trimester of pregnancy is suggested, as this is when changes in iron levels are the greatest and when there may be a need for targeted management. Dietary patterns do not seem to play a significant role in ID and ID anaemia in pregnant women, but this should be investigated more precisely.

## Figures and Tables

**Figure 1 medicina-58-00955-f001:**
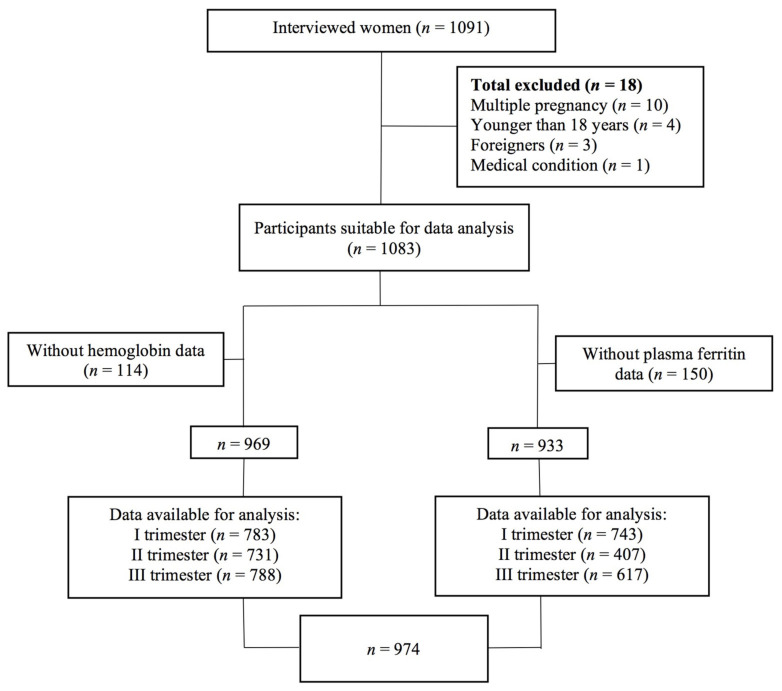
Schematic presentation of the sampling procedure and availability of haematologic data.

**Table 1 medicina-58-00955-t001:** Levels of Hb and SF in the three trimesters of pregnancy.

Indicators	Hb, g/L			SF, µg/L		
	I Trimester (*n* = 783)	II Trimester (*n* = 731)	III Trimester (*n* = 788)	I Trimester (*n* = 743)	II Trimester (*n* = 407)	III Trimester (*n* = 617)
Median	127.0	117.0	115.0	31.8	16.1	11.3
IQR (Q1 to Q3)	121.0–132.0	111.0–122.0	109.0–121.0	17.6–53.1	10.0–27.4	7.8–17.6
Range	95.0 to 184.0	84.0 to 270.0	76.0 to 150.0	4.2 to 187	1.1 to 157.0	2.3 to 201

**Table 2 medicina-58-00955-t002:** ID and ID anaemia data according to WHO and UK criteria.

	UK Recommendations					
Indicators	ID anaemia			ID		
	I trimester (*n* = 783)	II trimester (*n* = 731)	III trimester (*n* = 788)	I trimester (*n* = 743)	II trimester (*n* = 407)	III trimester (*n* = 617)
*n*	22	58	96	347	318	566
%	2.8	7.9	12.2	46.7	78.1	91.7
95% CI	1.9 to 4.2	6.2 to 10.1	10.1 to 14.7	43.1 to 50.3	73.9 to 81.9	89.3 to 93.7
	WHO					
Indicators	ID anaemia			ID		
	I trimester (*n* = 783)	II trimester (*n* = 731)	III trimester (*n* = 788)	I trimester (*n* = 743)	II trimester (*n* = 407)	III trimester (*n* = 617)
*n*	22	58	213	136	184	406
%	2.8	7.9	27.0	18.3	45.2	65.8
95% CI	1.9 to 4.2	6.2 to 10.1	24.1 to 30.2	15.7 to 21.3	40.4 to 50.1	62.0 to 69.4

**Table 3 medicina-58-00955-t003:** Maternal characteristics and the prevalence of anaemia by demographics or BMI.

Demographic Characteristics or BMI	I Trimester (*n* = 783)		II Trimester (*n* = 731)		III Trimester (*n* = 788)	
	Anaemic *	Non-Anaemic	Anaemic *	Non-Anaemic	Anaemic *	Non-Anaemic
	*n* (%)	*n* (%)	*n* (%)	*n* (%)	*n* (%)	*n* (%)
Total	22 (2.8)	762 (97.2)	58 (7.9)	674 (92.1)	213 (27.0)	576 (73.0)
Age group						
<20	0 (0.0)	10 (100.0)	2 (20.0)	8 (80.0)	6 (54.5)	5 (45.5)
20–24	5 (3.4)	144 (96.6)	13 (9.8)	120 (90.2)	43 (38.7)	107 (71.3)
25–29	4 (2.1)	186 (97.9)	11 (6.2)	166 (93.8)	51 (25.8)	146 (74.2)
30–34	7 (2.5)	268 (97.5)	21 (7.8)	248 (92.2)	77 (28.3)	195 (71.7)
≥35	6 (3.8)	153 (96.2)	11 (7.7)	131 (92.3)	36 (22.8)	122 (77.2)
BMI before pregnancy						
<18.5	0 (0.0)	32 (100.0)	2 (6.3)	29 (93.7)	11 (32.4)	22 (67.6)
18.5–24.9	16 (3.4)	459 (96.6)	4 (9.6)	397 (90.4)	120 (25.5)	351 (74.5)
25.0–29.9	4 (2.2)	180 (97.8)	10 (6.1)	154 (93.9)	60 (33.5)	119 (66.5)
≥30.0	1 (1.6)	62 (98.4)	2 (3.1)	63 (96.9)	15 (19.7)	61 (80.3)
No data **	27 (3.4)		32 (4.4)		29 (3.7)	
Education						
Primary or unfinished secondary	3 (6.5)	43 (93.5)	4 (9.1)	40 (90.9)	15 (31.9)	32 (68.1)
General secondary	2 (2.0)	97 (98.0)	8 (8.3)	88 (91.7)	28 (27.5)	74 (72.5)
Vocational secondary	6 (3.8)	154 (96.2)	13 (8.7)	137 (91.3)	48 (31.6)	104 (68.4)
Higher or unfinished higher	11 (2.3)	467 (97.7)	33 (7.5)	408 (92.5)	122 (25.0)	365 (75.0)
Marital status						
Married	8 (1.6)	496 (98.4)	33 (7.0)	440 (93.0)	122 (24.0)	395 (76.0)
Living in partnership	13 (6.2)	197 (93.8)	20 (10.2)	176 (89.8)	63 (30.7)	142 (69.3)
Single	0 (0.0)	59 (100.0)	4 (7.5)	49 (92.5)	24 (36.9)	41 (63.1)
Divorced or live separately, widow	1 (11.1)	8 (88.9)	1 (12.5)	7 (87.5)	3 (30.0)	7 (70)
No data **	1 (0.1)		1 (0.1)		1 (0.1)	
Nationality						
Latvian	17 (3.0)	541 (97.0)	38 (7.4)	476 (92.6)	151 (26.9)	410 (73.1)
Russian	4 (2.3)	170 (97.7)	12 (7.0)	159 (93.0)	41 (22.9)	138 (77.1)
Other	1 (2.0)	50 (98.0)	8 (17.4)	38 (82.6)	21 (43.8)	27 (56.3)
Residence						
Riga	12 (3.5)	327 (96.5)	34 (10.5)	291 (89.5)	90 (26.7)	247 (73.3)
Another city in Latvia	7 (2.1)	320 (97.9)	15 (5.0)	284 (95.0)	88 (26.9)	238 (73.1)
Village/rural area	3 (2.6)	112 (97.4)	9 (8.6)	96 (91.4)	35 (28.5)	88 (71.5)
No data **	2 (0.3)		2 (0.3)		2 (0.3)	
Number of births						
1	10 (2.7)	367 (97.3)	27 (7.4)	339 (92.6)	110 (27.7)	287 (72.3)
2	8 (3.1)	249 (96.9)	18 (7.8)	212 (92.2)	62 (24.4)	192 (75.6)
≥3	4 (2.7)	145 (97.3)	13 (9.6)	122 (90.4)	41 (29.9)	96 (70.1)

* According to the classification of the World Health Organization (WHO/UNICEF/UNO.IDA, 1998). ** From the total number.

**Table 4 medicina-58-00955-t004:** The odds ratio of ID anaemia in the third trimester.

Demographic Characteristics or BMI	OR (95% CI)	B *	S.E. *	Wald *	Exp(B) *	95% C.I.for EXP(B) *	*p* *
Marital status							
Married	(Ref)						
Living in partnership	1.40 (0.98 to 2.01)	0.308	0.190	2.628	1.36	0.94 to 1.98	0.105
Single	1.85 (1.07 to 3.18)	0.626	0.287	4.746	1.87	1.07 to 3.29	0.029
Divorced or live separately, widow	1.35 (0.34 to 5.31)	0.016	0.815	0.000	1.02	0.21 to 5.01	0.984
Nationality							
Latvian	(Ref)						
Russian	0.81 (0.54 to 1.20)	−0.142	0.211	0.455	0.87	0.57 to 1.31	0.500
Other	2.11 (1.16 to 3.85)	0.737	0.314	5.497	2.09	1.13 to 3.87	0.019
BMI before pregnancy							
18.5–24.9	(Ref)						
<18.5	1.46 (0.69 to 3.11)	0.336	0.387	0.753	1.40	0.66 to 2.99	0.386
25.0–29.9	1.48 (1.02 to 2.14)	0.381	0.194	3.881	1.46	1.00 to 2.14	0.049
>30	0.72 (0.39 to 1.31)	−0.283	0.310	0.833	0.75	0.411 to 1.38	0.362

* adjusted.

**Table 5 medicina-58-00955-t005:** Dietary intake according to the prevalence of anaemia in the third trimester of pregnancy (*n* = 788).

	All *n* (%)	Anaemic *n* (%)	Non-Anaemic *n* (%)	*p*
Meat (at least 3 portions a week)				
0 portions	20 (2.5)	7 (35.0)	13 (65.0)	0.551
<3 portions	97 (12.3)	29 (29.9)	69 (70.1)	
3+ portions	671 (85.2)	177 (26.4)	494 (73.6)	
Fish (at least 3 portions a week) *				
0 portions	125 (15.9)	38 (30.4)	87 (69.6)	0.326
<2 portions	540 (68.6)	148 (27.4)	392 (72.6)	
2+ portions	122 (15.5)	27 (22.1)	95 (77.9)	
Dairy products (3–4 times a day) **				
0 portions	8 (1.0)	3 (37.5)	5 (62.5)	0.292
<3 portions	494 (62.9)	143 (28.9)	351 (71.1)	
3 to 4 portions	160 (20.4)	35 (21.9)	125 (78.1)	
>4 portion	123 (15.7)	31 (25.2)	92 (74.8)	
Vegetables (at least 3 portions a day)				
<3 portions	778 (98.7)	212 (27.2)	566 (72.8)	0.302
3+ portions	10 (1.3)	1 (10.0)	9 (90.0)	
Fruits (at least 2 portions a day)				
<2 portions	686 (87.1)	182 (26.5)	504 (73.5)	0.413
2+ portions	103 (12.9)	31 (30.4)	71 (69.6)	
Coffee (every day) *				
Never	140 (17.8)	44 (31.4)	96 (68.6)	0.339
Sometimes	233 (29.6)	64 (27.5)	169 (72.5)	
Every day	414 (52.6)	104 (25.1)	310 (74.9)	

* *n* = 787, ** *n* = 785, *p* value—chi square.

## Data Availability

The data presented in this study are available on request from the corresponding author.

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
