# Peer review of "Iron Status in Pregnant Women in Latvia: An Epidemiological, Cross-Sectional, Multicenter Study According to WHO and UK Criteria"

_medicina, 2022, doi:10.3390/medicina58070955_

Round 1

Reviewer 1 Report

SECTIONAL STUDY

The aim of this study is to investigate the prevalence of iron deficiency, iron deficiency anaemia, and related dietary habits in pregnant women in Latvia. Its main contribution is providing with novel data about this issue in Latvia, since there are not enough reports of the problem in this context.

This paper is very interesting and considers an important public health problem. However, there are some limitations that need to be addressed in order to improve its clarity, beginning from its aim, main outcome, secondary outcomes, statistical analyses and the way the results are presented.

In the next pages, I include specific comments about these issues.

ABSTRACT

1.      Lines 15-16. Why were postpartum women included, since you are interested in pregnant women? They are in a different condition. If you decide to include postpartum women, I suggest to differentiate how many pregnant and how many postpartum women were included.

2.      Line 17: Consider rewording and English style in the phrase: “urbanization of women of childbearing age…”.

3.      Lines 19-20: It is confusing for me the stratification principal you mentioned in line 17, and reading that you reviewed medical files and patients of medical institutions… I do not see the relationship. Also, how did you get access to these institutions? How many were selected? Which criteria was followed to select them?

4.      Lines 20-22. These statistical analyses are not related to the aim of the study mentioned in lines 13-14. I suggest changing them to reflect what you actually did in your study. Also, how did you assess dietary patterns or dietary habits (they are not the same).

5.      Criteria for iron deficiency and anaemia should be stated in the Methods section, instead of the Results section.

6.      Lines 26-27. Consider rewording and check the English style in the phrase: “Single women had OR 1.85 (95% CI 1.07 to 3.18) times the odds of being anaemic then married women”.

7.      Conclusions: You do not mention the results for the performed associations. What do you mean with the phrase: “Management and supplementation should be performed in a timely and more targeted manner”. Was this assessed? In the Methods section you mentioned something about supplementation, but in the Results section, you did not mention anything about this. Is that really the problem behind iron deficiency and anaemia? Note: the conclusions in the manuscript are clearer and more objective than those presented in the Abstract section.

INTRODUCTION

1.      Line 38: Consider changing “in” for “during” in the phrase: “Maternal ID with or without anaemia in pregnancy”.

2.      Line 52: If you use the “IDA” acronym for “iron deficiency anaemia”, it should always be used. I think that it is not necessary, since ID anaemia is better understood. Check that for the whole paper.

3.      Lines 69-71: this paragraph is more related to the paragraph in lines 56-63. Consider integrating the information in the mentioned paragraph.

4.      Lines 72-80. Consider merging both paragraphs.

5.      Lines 85-85. Consider merging both paragraphs. In addition to this, I have some doubts about the sequence of the information and how the aim of the study is established at the end: In the previous paragraph you were talking about iron supplementation, and then, you focus in the lack of information about ID and IDA, when previously, you mentioned a WHO report (reference 15, which I could not retrieve) that had already established anaemia’s prevalence in pregnant women. Also, you mentioned another report (reference 16, which I could not retrieve either) that stated that this prevalence has not changed over the years. So, what is the novelty of your study? You say that information is scarce, but if those reports are the only information available, you should clearly state this. Moreover, how is this related to eating habits and iron supplementation? This also needs to be addressed in a clearer manner, to have coherence with the previous paragraph.

6.      Since I could not retrieve references 15 and 16, I checked for this information in the World Health Organization, Global Health Observatory Data Repository/World Health Statistics, available at https://data.worldbank.org/indicator/SH.PRG.ANEM?locations=LV (it might be the same reference you mentioned). It could be interesting to describe in your paper how the anaemia prevalence decreased drastically between the years 2000 and 2010 and then, how it has been stable since then, and why both changes (the decrease and the stabilization) have been observed (I have a doubt: the used criteria changed over these years?).

METHODS

1.      Line 94: Why postpartum women were included?

2.      Lines 95-96: How many institutions were included? I think that this is a multicentre, cross-sectional study; consider adding this to your study design.

3.      Lines 97-102: Consider merging and changing redaction style, so inclusion and exclusion criteria do not look like bullet points.

4.      Lines 103-104: You mention sample size calculation, but you do not mention the used criteria method for sample size calculation, nor the obtained sample size, nor the sampling method. Also, this is not reflected in the Abstract section, and vice versa.

5.      Lines 105-111: Consider changing the way frequencies are expressed; for example: instead of “18/1091”, say “18 subjects from 1091” or something similar (I did not understand the “/” symbol at first).

6.      Lines 108-109: Separate how many pregnant and postpartum women were included.

7.      Line 109: You should state that participants were interviewed to obtain different nutrition and health data, which are described in the “Collection data” section.

8.      Lines 115-116: I suggest including a table which evidences what you mention in these lines (it could be an Appendix).

9.      Figure 1: This figure is from Word and spelling errors are detected; consider changing this. Also, consider rewording in several places:

a.      The phrase “18 subjects dropped out” is incorrect, since they actually did not fulfil inclusion criteria.

b.      Delete the word “are” in the phrase: “Participants are suitable for data analysis”.

c.       Change the “N” for the “n” in patients with haemoglobin data.

10.  Line 118-119: It seems like you used a specific questionnaire for dietary habits and influencing factors assessment; if so, I suggest capitalizing the name of the questionnaire and using quotation marks. If a single questionnaire was used, change “were” for “was” in the phrase “were adapted from”. If neither of these observations are applicable, consider rewording these lines.

11.  Line 11: What does the “CDPC” acronym stands for?

12.  Line 120: Add the word “were” before the phrase “based on WHO expert recommendations”.

13.  Line 122: What do you mean with the phrase: “the adequate formulation of answers”?

14.  Line 125: Which anthropometric data was obtained?

15.  Lines 126-127: Eating habits (as you mention them) were assessed through a food frequency consumption questionnaire or through some questions? Is this a validated tool? You should explain this with more detail.

16.  Lines 128-129: Consider integrating information; for example: “Data on iron status (Hb and SF) were obtained from each participant’s medical records”.

17.  Lines 128-136: Consider merging both paragraphs, since they are closely related.

18.  Line 139: You stopped using the “Hb” and “SF” acronyms. In general, I think that these acronyms are not necessary.

19.  Lines 137-140: It replicates information from lines 130-136.

20.  Line 141: You previously mentioned that the questionnaire included anthropometric data, but weight was obtained from medical records. Did you perform another anthropometric measurement? Please, clarify this.

21.  Line 143: What do you mean with the phrase “and controlling the weight of women at each visit”? Consider rewording.

22.  Lines 146-157: Check the format and redaction style; this can be simplified.

23.  Line 168: Consider using a “semicolon” instead of a “colon” after the word “obtained”.

24.  Lines 169-170: Mean (SD) are not mentioned (you mention this in lines 171-172).

25.  Line 170: Why did Shapiro-Wilk test was used for normal distribution assessment (instead of Kolmogorov-Smirnov test)?

26.  Lines 171-172: Here you state that you used mean (SD). Please, check for coherence with respect to lines 169 and 170.

27.  Lines 172-182: Consider including this information in another separate paragraph.

28.  Line 173: Hb and SF were obtained from medical records, right? As it is written, it seems like you analysed blood samples.

29.  Lines 174-175: Check the redaction style in the phrase: “between women without ID and women with IDA”. I think you wanted to say “between women with and without IDA”.

30.  Line 179: Add the word “as” before the words “statistically significant”. In addition, information about p values should be placed after describing the used statistical analysis package.

31.  In general, it is not clear to me when the comparisons will be performed: only for the last trimester? Between trimesters? 

RESULTS

1.      I think that information is not presented in the same order than that described in the Methods section. Try to better organize your ideas: which is the main outcome? According to your title, ID and IDA is the more important outcome (in other words, the 3.2 subheading). Which are the secondary outcomes? According to that, organize your statistical analysis (in the Methods section) and your Results’ subheadings (which need to be improved).

2.      You should also be careful with the term “range”, because in statistics this is the difference between the lowest and the highest values (and that is not what you are presenting).

3.      Why are the comparative analysis performed only according to IDA classification? Did you perform this analysis according to ID classification? This is confusing.

4.      Also, you need to clearly state which statistical analysis were performed in each one of the tables.

5.      Line 186: Are you sure that the BMI value is for the first trimester? I would think that this is the prenatal BMI value. Please confirm.

6.      Line 188: Consider rewording the phrase “maternal demographics and characteristics”. I think that it should say: “maternal demographic characteristics and prenatal BMI classification”.

7.      Lines 189-190. Consider rewording when comparing between women with and without anaemia.

8.      Table 1. I have a few notes for this table:

a.      Consider deleting the p value columns and adding table notes for these, in order to simplify the information and improving its clarity.

b.      Check if you meet chi-squares’ application conditions (especially, the expected frequency of at least 5). Also, since it is not a 2 x 2 table, it is important to explain which groups are different when significant differences are detected (you may need extra chi-square tests).

c.       Check data since I detected some errors; for example: 100% of women < 20 years were not anaemic, which in theory corresponds to 15 subjects (not 10). However, since you have different samples sizes per trimester, I would suggest to exclude missing data or include table notes to differentiate sample sizes, in the whole table. I see the “No data **” note, but this is not always present or the information is not always coherent.

d.      Did you perform analysis of changes in the proportion of anaemic women between trimesters (changes across time)? I think that this would indeed change. You could perform McNemar’s tests, but this is only for 2 x 2 tables, so several tests should be performed. You could also look for more suitable tests.

e.      Line 193: Add a “dot” at the end of the note.

9.      Line 195: delete the “-” sign.

10.  Lines 195-197. Redaction style is not clear enough. Consider rewording.

11.  Table 2. This table needs to be improved. There is lacking information about why you chose those variables for the analysis (because you have not included information about dietary parameters, for example) or why some of the whole variables were used as confounders. You also need to add more explanatory note tables. BMI is not a demographic characteristic, from my point of view. What happens if you group subjects with prenatal BMI? Is there a change in the association?

12.  Table 3. Include mean and SD data. It is mentioned in the text and in the Discussion section.

13.  Lines 202-222. Information should be better described and organized (it is too fragmented).

14.  Lines 223-225. This should be in the Methods section.

15.  Lines 226-138: Information should be better described and organized. I suggest including a table with the performed correlations to better understand the message you are trying to transmit.

16.  Lines 239-253: This should be described before the correlations and the associations analyses.

17.  Line 253: Consider rewording “by the prevalence” to “according to the prevalence”.

18.  Line 240: This should be stated in the Methods section.

19.  Lines 241-249. Redaction style should be more fluent. Please revise this.

20.  Lines 250-251: Consider changing the text order: “There were no statically significant differences in eating habits between participants with and without anaemia…”.

21.  Line 252: You should specify which are the sources of iron and which are the foods that influence iron absorption.

22.  Table 4.

a.      Consider changing the column title for Hb; I would name it as you did in the previous tables: anaemic, non-anaemic women.

b.      Total n is missing in some variables.

c.       What does “None” mean?

DISCUSSION

1.      Line 256: Confirm the study’s aim.

2.      Line 262-264: Consider rewording the next phrase (it not clear): “but analysed while taking into account UK guidelines that are more recent and that gynaecologists and obstetricians in Latvia follow [20].”.

3.      Line 270: Consider deleting the word “Thus”.

4.      Line 288: What does “sTfR” stands for? Serum transferrin? It should be stated.

5.      Lines 288-289: Consider grouping information about ID prevalence according to SF levels (three countries use the same criteria, while only one uses another criteria).

6.      Line 291: Include a reference citation at the end of the sentence.

7.      Lines 292-293: Consider changing text order: “The mean haemoglobin and ferritin values were analysed separately in all trimesters to determine…”.

8.      Line 299: Move citation before the word “similar”.

9.      Line 306: You mention “other studies” but you include only one citation. Consider including more citations or change the redaction style.

10.  Line 319: A citation is needed when describing haem iron sources (meat and fish).

11.  Line 321: Consider rewording the phrase: “with its main sources being…”. Citation is needed at the end of the sentence. Now, not all the fruits or vegetables are good sources of Vitamin C; this is a potential limitation of the tool and this should be addressed in the Discussion.

12.  Line 323: A citation is needed when describing foods that inhibit iron absorption.

13.  Line 325: Change “was found” for “were found”.

14.  Line 328: You should also mention if the questionnaire is validated and how this could be a potential limitation.

15.  Line 334: A citation is needed. Other sources of limitation (including the comment about Vitamin C sources that I previously mentioned) are: 1) the way weight was recorded (weight technique nor equipments for its assessment are described, inter-observers’ measurement error is not known and, in some cases, data is absent or auto-reported weight is used instead); and 2) not knowing if the methods for Hb and SF measurement are similar between institutions.

REFERENCES

Check references according to the reference formatting guide and review that all Internet sources can be downloaded.

Reviewer 2 Report

Authors presented retrospective analysis on iron deficiency and related anemia in Latvian population of pregnant women. The criteria followed WHO and UK guidelines. It is a part of a larger study on pregnant population in Latvia as pointed out in Funding Policy. The presented data is giving more arguments to change the national policy in targeting deficient patients to improve pregnancy and birth outcomes in Latvia. However we cannot conclude from the presentation what impact if any has the current prevalence of ID/IDA on perinatal/birth/neonatal outcomes nor which criteria (WHO or UK) are better in predicting the pregnancy complications in studied population.

To improve the manuscript authors may want to change and implement:

1/ add term "related anemia by WHO and UK criteria" in the title

2/ explain which system is used: WHO or UK in Latvia and why

3/ give the total number of pregnancies/births in Latvia during the study period

5/ present the data on ID and IDA in extra table with separate columns for WHO and UK criteria respectively

6/ provide the basic data on birth outcomes in studied groups to check if there are differences between anemic and non-anemic groups and ideally between WHO or UK criteria separately vs non-anemic.

7/ in discussion to compare the findings with available data from other Baltic sea countries e.g. Lithuania, Estonia, Finland, Sweden to check if there are significant differences

8/ correct references according to the journal rules e.g by removing quotation marks where not necessary.

Author Response

This manuscript is a resubmission of an earlier submission. The following is a list of the peer review reports and author responses from that submission.